# FAANG Stocks, Gold, and Islamic Equity: Implications for Portfolio Management during COVID-19

**Kashif Saleem** [1,*], **Osama AlHares** [1], **Haroon Khan** [1] and **Omar Farooq** [2]

[1] Faculty of Business, University of Wollongong in Dubai, Dubai P.O. Box 20183, United Arab Emirates
[2] School of Business, ADA University, Baku P.O. Box AZ1008, Azerbaijan
[*] Correspondence: kashifsaleem@uowdubai.ac.ae; Tel.: +971-4-2781947

**Abstract:** During the COVID-19 pandemic, technology stocks, such as FAANG stocks (Facebook, Amazon, Apple, Netflix, and Google), attracted the attention of global investors due to the vast use of technology in daily business. However, technology stocks are generally considered risky stocks; hence, efficient risk management is required to construct an optimal portfolio. In this study, we investigate the volatility spillovers and dynamic conditional correlations among the daily returns of FAANG company stocks, gold, and sharia-compliant equity to construct the optimal portfolio weights and hedge ratios during the COVID-19 pandemic period by utilizing a multivariate GARCH framework. The dynamic conditional correlations reveal that both gold and sharia-compliant equities exhibit lower correlations with FAANG stocks during the COVID-19 pandemic, implying opportunities for portfolio diversification. The findings indicate that gold and shariah-compliant equity are good candidates to hedge FAANG stocks. These findings are highly relevant for international investors, asset managers, hedgers, and portfolio managers.

**Keywords:** COVID-19; FAANG stocks; Islamic Equity; portfolio management

## 1. Introduction

The aim of this paper is to document the hedging prospects of gold and shariah-compliant equities during the COVID-19 pandemic. We believe that it is particularly important to understand the hedging prospects of these assets during the COVID-19 pandemic because this period was characterized by one of the worst global recessions of the recent times (Shen et al. 2020). In addition to the impact on real markets, the COVID-19 pandemic also affected the stock markets (Al-Awadhi et al. 2020; Ashraf 2020; Baker et al. 2020; Ramelli and Wagner 2020; Zhang et al. 2020). Bossman et al. (2022) noted that conventional stocks were prone to more volatilities than Islamic stocks during market turbulence. This uncertainty in stock markets indicates a challenging situation, which calls for the deployment of effective and aggressive risk management strategies and the selection of asset classes with low or negative correlations during the pandemic period. Investors, hedgers, and asset managers, therefore, should be interested in identifying the asset classes that can reduce their exposure to risk.

The motivation behind understanding the hedging prospects of gold and shariah-compliant equities is based on the assumption that these assets possess a lower risk. A significant amount of the prior literature shows that gold is a good portfolio diversifier during uncertain times (Gürgün and Ünalmış 2014; Beckmann et al. 2015; Sherman 1982; Jaffe 1989; Baur and Lucey 2010; Ciner et al. 2013; Bekiros et al. 2017). AlKhazalia et al. (2021), for example, showed that a portfolio consisting of gold had lower risk than a portfolio that excluded gold. Likewise, Baur and McDermott (2016), among others, showed that gold acted as a strong candidate for optimal portfolio construction during 11 September 2001 and the Lehman insolvency in September 2008. This strand of the literature argues that the ability of gold to reduce portfolio risk depends on a number of factors. For instance,

Baur and McDermott (2016) noted that behavioral biases linked with gold's history as a currency or a store of value was the main reason behind safe haven properties of gold. Another important reason behind the ability of gold to reduce portfolio risk is based on the assumption that when other assets do not do well, investors turn to precious metals, such as gold, to safeguard themselves against adverse performance of other investments. As a result, the demand for gold increases, which has a positive impact on the prices of gold and leads to a lower variation of portfolios that include gold. These studies indicate that investors should include gold in their portfolios to maximize their expected returns, which is also supported by Yan and Garcia (2017) who studied the effectiveness of commodities in an optimal portfolio.

Similar to gold, shariah-compliant equities also possess characteristics that make them good candidates for acting as the safe havens during market downturns (Mirza et al. 2022; Saiti et al. 2014; Rizvi et al. 2015; Abbes and Trichilli 2015; Al-Zoubi and Maghyereh 2007; Milly and Sultan 2012). Ashraf et al. (2022), for example, reported that shariah-compliant equities outperformed their conventional counterparts in the first quarter of 2020, the peak of the COVID-19 pandemic. They documented that shariah-compliant equities provided hedging benefits during extreme market downfalls. Bossman et al. (2022) also documented the diversification opportunities between Islamic and conventional equities in the short- and mid-term periods of the COVID-19 pandemic. Ahmed and Farooq (2018) came to a similar conclusion, when they reported that the degree of shariah-compliance was a significant determinant of risk. They showed that the more compliant a portfolio was, the lower the risk of the portfolio was. Arif et al. (2021) also showed that shariah-compliant equities emerged as a robust safe-haven asset for the G7 stock markets during the pandemic crisis. Several arguments have been put forward for this relationship. Farooq and AbdelBari (2015), for example, argued that shariah-compliant equities had lower information asymmetries relative to noncompliant equities. Farooq and Ahmed (2022) documented lower information asymmetries of shariah-compliant equities due to their financial characteristics. They noted that lower levels of debt, cash, and account receivables associated with shariah-compliant equities resulted in a better information environment. The better information environment of shariah-compliant equities is also attributed to the increased scrutiny that these equities generate from investors and regulators. Haseeb et al. (2022) noted that managers in shariah-compliant equities pay close attention to not being caught in immoral actions. Therefore, they tend to manage their firms more ethically.

In this paper, we extend the literature on the safe haven properties of gold and sharia-compliant equities by documenting their impact on portfolios consisting of FAANG (Facebook, Amazon, Apple, Netflix, and Alphabet) stocks. These stocks traditionally generate lot of interest from stock market participants and constitute a large proportion of the S&P 500 index and NASDAQ index. The paper explores the diversification opportunities by constructing an optimal portfolio and offering a hedging strategy by computing the hedge ratios for risk management during the COVID-19 pandemic period. First, we investigate the return and volatility spillover effects among the daily closing prices of three asset classes, namely the FAANG stocks representing the technology sector and tracked by the NYSE FANG + TM Index, shariah-compliant equity measured by FTSE All-World Shariah Index, and gold spot prices using the VAR–GARCH BEKK representation proposed by Engle and Kroner (1995). More recently, several studies have utilized the Markov switching process to capture the volatility dynamics (Zheng and Zuo 2013; Balcilar et al. 2013) and to forecast volatility (Ardia et al. 2018); however, to examine the hedging and diversification opportunities, the Dynamic Conditional Correlation (DCC) framework proposed by Engle (2002) remains the standard methodology, which we utilize in this study. Our main findings indicate that the time-varying correlations are low compared to the static correlation; therefore, it is important to employ the DCC framework to calculate the time-varying correlations to avoid misleading conclusions. Moreover, we find that gold and shariah-compliant equity are both good candidates to hedge FAANG stocks. For example, a USD 1000 investment in FAANG stocks can be hedged with a short position in gold

with USD 490 investment, whereas only USD 20 investment in shariah-compliant equity is necessary. The optimal weight for the FAANG/gold portfolio is 0.14, whereas the optimal weight for the FAANG/shariah-compliant portfolio is 0.23. Overall, these results imply that gold and shariah-compliant equity can provide portfolio diversification benefits. These findings are relevant for investors, hedgers, and portfolio managers.

We further contribute to the growing body of literature on the impact of COVID-19 on stock prices, specifically from the portfolio diversification viewpoint. As per the authors' best knowledge, very few studies from the portfolio diversification viewpoint during the COVID-19 are available in the existing literature; the exceptions include, for example, Corbet et al. (2020). Our paper fills this gap by presenting new evidence on the optimal hedge and diversification opportunities for FAANG stocks investors with the well-known diversifiers, gold and shariah-compliant equity.

In the next section, the study outlines the source of data used to generate the results, while Section 3 provides the methodology. In Section 4, we discuss the empirical results, while Section 5 summarizes the main findings from the study.

## 2. Data

The data for this study were retrieved from the Thomson Reuters database, which comprises the daily closing prices of the NYSE FANG+TM Index (FAANG), the FTSE All-World Shariah Index (shariah-compliant equity) and gold spot prices (gold). The NYSE FANG+TM Index traces ten actively traded technology stocks. It is an equally weighted Index of Facebook, Amazon, Apple, Tesla, Netflix, NVidia, Twitter, Alibaba, Baidu, and Alphabet. The FTSE Sharia Global Equity index covers the stocks of all sharia-compliant companies in both developed and emerging markets. Our investigation period covered the ongoing COVID-19 pandemic, starting from 1 January 2020 and ending on 25 September 2020, comprising 186 daily observations. As a first step, we calculated the compounded daily returns $rt = 100 \times \ln \frac{P_t}{P_{t-1}}$, by using the closing prices ($pt$) of all assets under investigation. The descriptive statistics for the return series of FAANG, gold, and shariah-compliant equity are reported in Table 1. For each asset class, the mean and median values were positive. For each asset class, the standard deviation was larger than the mean value. For each asset class, Student's t test showed that the mean was statistically insignificant from zero. Each asset class showed a small amount of negative skewness and a moderate amount of kurtosis, and the returns were not normally distributed. In addition, the summary statistics for unconditional correlations revealed that there was a moderate positive correlation between FAANG, gold, and shariah-compliant equity; however, it was less than 0.50, which provides a large amount of room for diversification. It is clearly evident from the time series graphs, Figure 1, of the squared daily returns of the FAANG stocks, gold, and shariah-compliant equity that the volatility has changed across time. All asset classes showed volatility clustering between March 2020 and April 2020. Moreover, the FAANG stocks exhibited some large jumps in volatility in March 2020. The information presented in Figure 1 shows volatility clustering and the development of the three asset classes under investigation.

**Table 1.** Summary statistics for daily returns.

|  | FAANG | GOLD | SHARIAH |
|---|---|---|---|
| Mean | 0.2634 | 0.1065 | 0.0136 |
| Median | 0.5932 | 0.1497 | 0.0904 |
| Maximum | 7.2975 | 5.1334 | 7.8940 |
| Minimum | −14.0490 | −5.2646 | −9.8442 |
| Variance | 7.7641 | 1.7588 | 3.4261 |

**Table 1.** *Cont.*

|  | **FAANG** | **GOLD** | **SHARIAH** |
|---|---|---|---|
| Skewness | −1.2203 | −0.5813 | −1.2267 |
| Kurtosis | 4.5958 | 3.8736 | 8.1364 |
| Student's t | 1.2859 | 1.0924 | 0.0997 |
| Jarque–Bera | 208.7228 | 126.0808 | 556.6915 |
| Probability | 0.0000 | 0.0000 | 0.0000 |
| Observations | 185.0000 | 185.0000 | 185.0000 |

The table presents the descriptive statistics of the return series of FAANG stocks, gold, and sharia-compliant stocks, including the mean, variance, skewness, and kurtosis. The Jarque–Bera test for normality is based on the skewness and excess kurtosis.

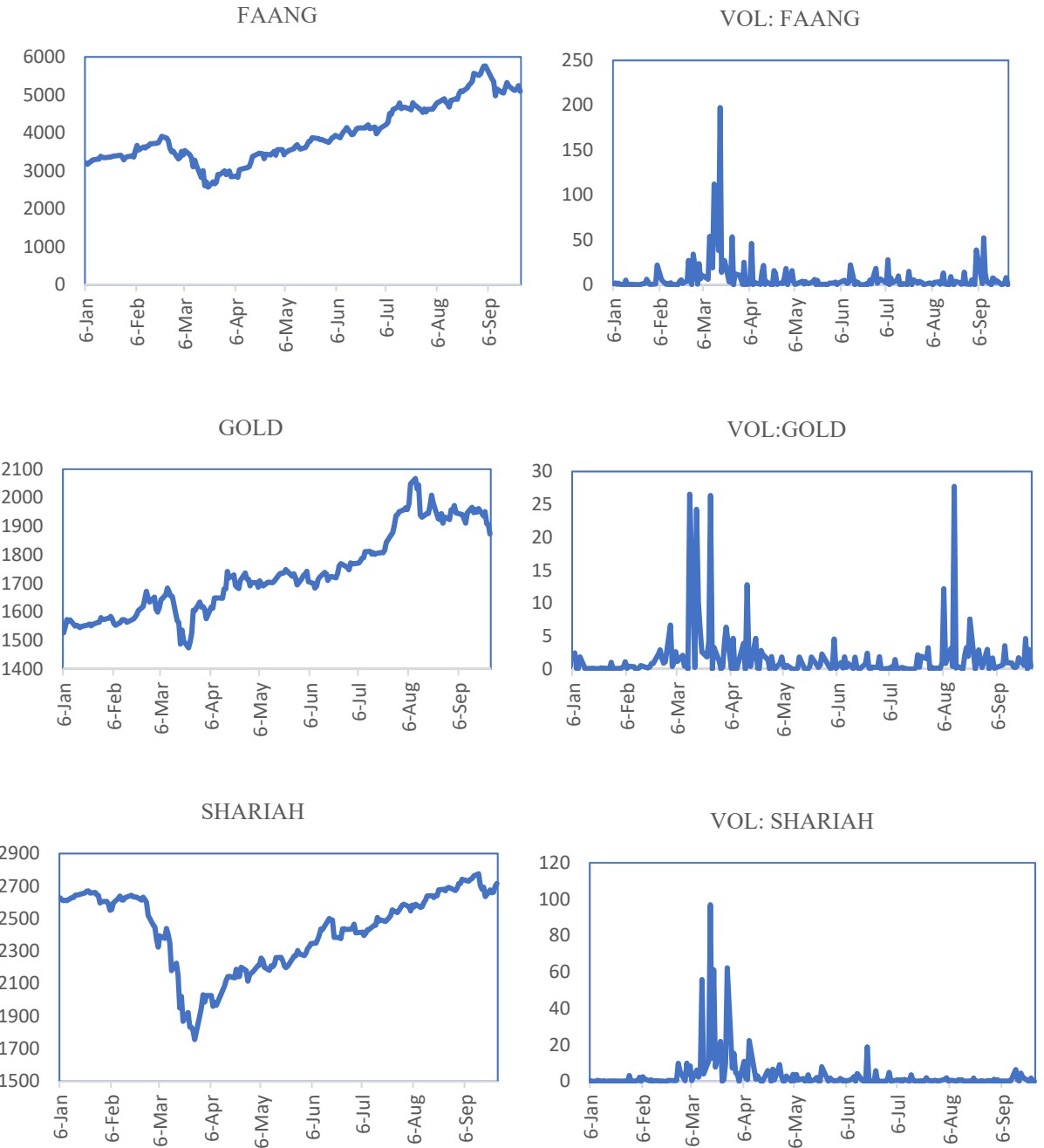

**Figure 1.** Time-varying hedge ratios computed from the DCC model.

### 3. Empirical Framework

The Autogressive Conditional Heteroscedasticity (ARCH) process presented by Engle (1982) and the generalized ARCH (GARCH) process introduced by Bollerslev (1986) are considered the standard methods to model the volatility of stock returns in a univariate setting. However, to investigate the volatility spillovers among different classes of assets, a multivariate GARCH procedure is preferred. As a first attempt, Bollerslev et al. (1988), among others, proposed a diagonal VECH method to study the conditional covariances in the bond and stock market in the USA; however, the model failed to ensure the positive definiteness of the conditional variance matrix, which is one of the requirements to construct a multivariate model. Moreover, the diagonal VECH model was not able to capture the cross-equation effects. A substantial amount of literature has been published around multivariate GARCH models over the last couple of decades. Engle and Kroner (1995) proposed multivariate GARCH–BEKK (Baba, Engle, Kraft, and Kroner) parameterization, which overcame most of the problems inherited in the early models. However, the BEKK model only satisfied the hypothesis of constant correlation instead of the dynamic conditional correlations among different asset classes. Engle (2002) proposed a dynamic conditional correlation coefficient (DCC) model to overcome the BEKK limitations. In this study, we utilized both the GARCH–BEKK and GARCH–DCC model to investigate the volatility spillovers and conditional correlations among FAANG stocks, gold, and sharia-compliant equity.

*3.1. VAR–GARCH–BEKK Model*

To compute the return and volatility spillovers among FAANG stocks, gold, and shariah-compliant equity, we utilized a bivariate VAR–GARCH–BEKK model. The VAR model can accommodate each asset's returns and one period lagged returns of the other assets. Equation (1) represents the model.

$$
\begin{bmatrix} r_{1,t} \\ r_{2,t} \\ r_{3,t} \end{bmatrix} = \begin{bmatrix} a_{1,0} \\ a_{2,0} \end{bmatrix} + \begin{bmatrix} \beta_{1,1} & \beta_{1,2} \\ \beta_{2,1} & \beta_{2,2} \end{bmatrix} \begin{bmatrix} r_{1,t-p} \\ r_{2,t-p} \end{bmatrix} + \begin{bmatrix} \mu_{1,t} \\ \mu_{2,t} \end{bmatrix} \tag{1}
$$

$$
u_t | \Omega_{t-1} \sim N(0, H_t) \tag{2}
$$

The above bivariate construction enables the measurement of the effects of the mean series returns of one asset on its own lagged returns and those of the lagged returns of other assets. Here, $r_t$ represents the daily returns at time t for each asset. $\mu_t$ represents the error term for each asset at time t, with its corresponding conditional variance–covariance matrix $H_t$. $\Omega_{t-1}$ represents the information set available at time $t-1$. The parameter $\alpha$ represents the constant. The own mean spillovers and cross-mean spillovers are measured by the estimates of the matrix $\beta$ elements, the parameters of the vector autoregressive term.

Further, by adopting a multivariate GARCH–BEKK specification and following Engle and Kroner (1995), the conditional covariance matrix can be stated as:

$$
\begin{aligned}
H_t = C_0' C_0 &+ \begin{bmatrix} \gamma_{11} & \gamma_{12} \\ \gamma_{21} & \gamma_{22} \end{bmatrix}' \begin{bmatrix} \varepsilon_{1,t-1}^2 & \varepsilon_{1,t-1}, \varepsilon_{2,t-1} \\ \varepsilon_{1,t-1}, \varepsilon_{2,t-1} & \varepsilon_{2,t-1}^2 \end{bmatrix}' \begin{bmatrix} \gamma_{11} & \gamma_{12} \\ \gamma_{21} & \gamma_{22} \end{bmatrix} \\
&+ \begin{bmatrix} \delta_{11} & \delta_{12} \\ \delta_{21} & \delta_{22} \end{bmatrix}' H_{t-1} \begin{bmatrix} \delta_{11} & \delta_{12} \\ \delta_{21} & \delta_{22} \end{bmatrix}
\end{aligned} \tag{3}
$$

where the parameters for the variance equation are defined as $\gamma_{ii}$ and $\delta_{ii}$ to capture the short-term volatility persistence and long-term volatility persistence, respectively, whereas the parameters $\gamma_{ij}$ and $\delta_{ij}$ measure the cross-asset shock and volatility spillovers, respectively.

### 3.2. DCC–GARCH Model

To investigate the nature of the conditional correlations across the three series, we utilized a dynamic conditional correlation (DCC) model, proposed by Engle (2002) and Engle and Sheppard (2001). The DCC model may be expressed as follows:

$$H_t = D_t R_t D_t, \tag{4}$$

where $H_t$ represents the conditional covariance matrix representing the DCC specification with $D_t$ as the conditional volatility of the returns on each asset under investigation, defined as $D_t = \text{diag}(h_{iit}^{1/2} \ldots\ldots h_{kkt}^{1/2})$, while $R_t$ measures the conditional correlations among the three assets, defined as $R_t = (\text{diag}Q_t)^{-1/2} Q_t (\text{diag}Q_t)^{-1/2}$, where $Q_t = (1 - \alpha - \beta)\overline{Q} + \alpha\mu_{t-1}\mu'_{t-1} + \beta Q_{t-1}$ represents a symmetric positive definite matrix with $\mu_{it} = \varepsilon_{it} / \sqrt{h_{iit}}$, while, $\overline{Q}$ refers to the unconditional variance matrix of $\mu_t$, and $\alpha$ and $\beta$ are nonnegative scalar parameters satisfying $\alpha + \beta \prec 1$. Finally, the conditional correlation coefficient $\varrho_{ij}$ between two assets i and j can be expressed by the following equation:

$$\rho_{ij} = \frac{(1 - \alpha - \beta)\overline{q_{ij}} + \alpha\mu_{i,t-1}\mu_{j,t-1} + \beta q_{ij,t-1}}{\left\langle (1 - \alpha - \beta)\overline{q_{ii}} + \alpha\mu_{i,t-1}^2 + \beta q_{ii,t-1} \right\rangle^{1/2} \left\langle (1 - \alpha - \beta)\overline{q_{jj}} + \alpha\mu_{j,t-1}^2 + \beta q_{jj,t-1} \right\rangle^{1/2}} \tag{5}$$

As per Engle and Sheppard (2001) and Engle (2002), this model can be estimated with the quasi-maximum likelihood method (QMLE).

## 4. Empirical Results

### 4.1. Volatility Spillovers

Table 2 shows the volatility spillovers between the three assets, both in terms of short-term persistence (ARCH effect), represented by $\gamma_{ii}$, and long-term persistence (GARCH effect), represented by $\delta_{ii}$. Consistent with a typical GARCH framework, the results showed that for each asset, the estimated $\gamma_{ii}$ values were smaller than their respective estimated $\delta_{ii}$ values, which revealed that each asset was influenced by its own volatility in the long run rather than in the short run; however, only the estimated coefficients of the sharia-compliant equity, $\delta_{33}$, were statistically significant (*p*-value < 1%), indicating that the conditional volatility of sharia-compliant equity was predictable in the long run. Moreover, the findings from the BEKK model showed several cross-asset volatility spillover effects. For short-term volatility persistence, the findings indicated that a significant bidirectional relationship existed between FAANG and gold ($\gamma_{12}$, $\gamma_{21}$), as well as between FAANG and sharia-compliant equity ($\gamma_{13}$, $\gamma_{31}$). However, for long-term volatility persistence, we found no bidirectional relationship between FAANG and gold ($\delta_{12}$, $\delta_{21}$), as well as between FAANG and sharia-compliant equity ($\delta_{13}$, $\delta_{31}$). These results are important for portfolio managers and FAANG stocks investors to forecast the short-term volatility spillovers among FAANG stocks, gold, and sharia-compliant equity. Moreover, the findings of no long-term volatility persistence among FAANG stocks, gold, and sharia-compliant equity indicated the diversification opportunities. The diagnostic tests provided no evidence of serial correlation in both the standardized residuals and squared standardized residuals at the 1% significance level in the BEKK model.

**Table 2.** GARCH–BEKK parameter estimates of volatility spillovers among FAANG stocks, gold, and sharia-compliant equity.

| Panel A: Conditional ARCH effects | | | |
|---|---|---|---|
| | **Coeff** | **T-Stat** | **Signif** |
| $\gamma_{11}$ | 0.1869 | 1.5624 | 0.1182 |
| $\gamma_{12}$ | 0.1782 | 3.2690 | 0.0011 |
| $\gamma_{13}$ | 0.3007 | 3.9721 | 0.0001 |
| $\gamma_{21}$ | −0.8244 | −2.4283 | 0.0152 |
| $\gamma_{22}$ | 0.0216 | 0.0775 | 0.9382 |
| $\gamma_{23}$ | 0.0921 | 0.9165 | 0.3594 |
| $\gamma_{31}$ | −0.6406 | −3.8942 | 0.0001 |
| $\gamma_{32}$ | −0.1567 | −0.9488 | 0.3427 |
| $\gamma_{33}$ | −0.0990 | −1.2248 | 0.2207 |
| **Panel B: Conditional GARCH effects** | | | |
| $\delta_{11}$ | 0.1872 | 0.4219 | 0.6731 |
| $\delta_{12}$ | 0.2731 | 1.4546 | 0.1458 |
| $\delta_{13}$ | −0.0866 | −0.7057 | 0.4804 |
| $\delta_{21}$ | 0.6699 | 1.5367 | 0.1244 |
| $\delta_{22}$ | 0.5416 | 1.6391 | 0.1012 |
| $\delta_{23}$ | 0.2936 | 1.8650 | 0.0622 |
| $\delta_{31}$ | −0.1024 | −0.4372 | 0.6620 |
| $\delta_{32}$ | −0.1975 | −2.9480 | 0.0032 |
| $\delta_{33}$ | 0.7879 | 9.6696 | 0.0000 |
| **Panel C: Diagnostic tests** | | | |
| | **FAANG** | **GOLD** | **SHARIAH** |
| **Q20** | 16.3329 | 42.9652 | 27.0361 |
| *p* **Values** | 0.6958 | 0.0021 | 0.1343 |
| **Q²20** | 21.6294 | 24.7405 | 27.5196 |
| *p* **Values** | 0.3610 | 0.2115 | 0.1213 |

Note: The estimated parameters of the GARCH–BEKK model are presented in Panels A and B, 1 stands for FAANG, 2 represents gold, and 3 represents sharia-compliant equities. $\gamma_{ii}$ represents the ARCH effect, while $\delta_{ii}$ represents the GARCH effect, and Q-stat represents the standardized residuals and squared standardized residuals at lag 20.

### 4.2. Dynamic Conditional Correlations

The DCC model was utilized to construct the portfolio weights, hedge ratios, and dynamic conditional correlations among the three assets. The findings from the application of the DCC model showed that the dynamic conditional correlations ($\varrho_{12} = 0.248$, $\varrho_{13} = 0.026$, and $\varrho_{23} = 0.082$) were significantly lower than the constant conditional correlations ($\varrho_{12} = 0.37$, $\varrho_{13} = 0.26$, and $\varrho_{23} = 0.20$) as outlined in Table 3 and Figure 2. These findings emphasized the need to estimate the dynamic correlations to generate deeper and higher quality inferences. Estimation of the dynamic correlations showed that after December 2019, when the news of the pandemic spread, there was an upward trend in the dynamic correlations between the FAANG stocks and gold, which peaked at 80% by the last week of March. This specifies that there was a very narrow opportunity for portfolio diversification among these two assets in the early stage of pandemic. However, the dynamic conditional correlation started to decline after March 2020 and saw some episodes of negative correlations until August 2020, indicating the protentional of diversification benefits between the two classes of assets.

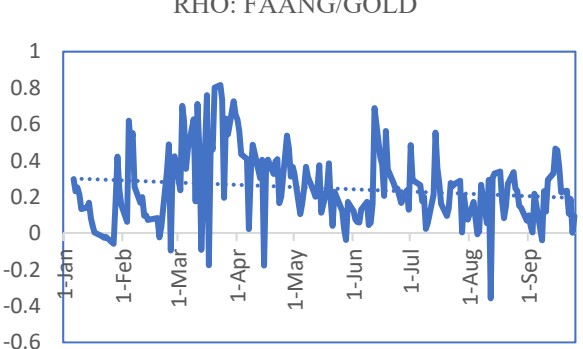

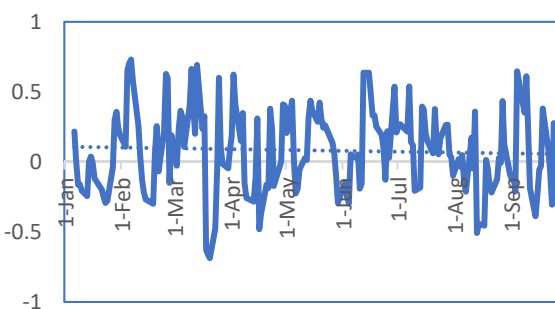

**Figure 2.** Time-varying conditional correlations.

**Table 3.** Static and dynamic conditional correlations.

| Panel A: Static correlations | | | |
|---|---|---|---|
| | FAANG | GOLD | SHARIAH |
| FAANG | 1.00 | 0.38 | 0.26 |
| GOLD | | 1.00 | 0.20 |
| SHARIAH | | | 1.00 |
| **Panel B: Dynamic conditional correlations- DCC framework** | | | |
| | FAANG | GOLD | SHARIAH |
| FAANG | 1.00 | 0.25 | 0.03 |
| GOLD | | 1.00 | 0.08 |
| SHARIAH | | | 1.00 |

The table presents the static and dynamic conditional correlations between FAANG stocks, gold, and sharia-compliant stocks. The dynamic conditional correlations are based on the multivariate GARCH–DCC framework.

Similar trends were observed in the dynamic correlations between the FAANG stocks and sharia-compliant equity with a peak correlation happening in June 2020 at 72% followed by a significant negative correlation (−70%) in August 2020, indicating the strong protential of diversification benefits between the two assets. Likewise, the time-varying conditional correlations between gold and shariah-compliant equity showed a similar pattern of negative correlation during the pandemic period. The time series plots generated by the dynamic conditional correlation model in Figure 2 gives further detailed information that supports the hypothesis that the overall correlations between the assets were lowest during the COVID-19 period, emphasizing, the superiority of the DCC model over a constant conditional correlation model.

The optimal portfolio weights, calculated by utilizing the DCC model, are presented in Table 4. As per our results, the optimal weight for the FAANG/gold portfolio was 0.14, signifying that in a USD 1 portfolio, comprised of FAANG stocks and gold, USD 0.86 should be invested in FAANG stocks and USD 0.14 in gold. Likewise, the optimal

weight for the FAANG/shariah-compliant equity portfolio specified that USD 0.77 should be invested in FAANG and USD 0.23 in shariah-compliant equity. The optimal weight for the shariah-compliant equity/gold portfolio showed that USD 0.56 should be invested in shariah-compliant equity and USD 0.44 in gold to realize the diversification benefits.

**Table 4.** Summary statistics of the hedge ratios (long/short) and portfolio weights.

| Panel A: Hedge ratios | | | | |
| --- | --- | --- | --- | --- |
| | Mean | St Dev | Min | Max |
| FAANG/GOLD | 0.49 | 0.44 | −1.30 | 2.01 |
| FAANG/SHARIAH | 0.02 | 0.48 | −1.97 | 1.50 |
| GOLD/FAANG | 0.14 | 0.13 | −0.10 | 0.77 |
| GOLD/SHARIAH | 0.05 | 0.27 | −0.93 | 0.61 |
| SHARIAH/FAANG | 0.02 | 0.15 | −0.74 | 0.44 |
| SHARIAH/GOLD | 0.12 | 0.37 | −1.07 | 1.12 |
| **Panel B: Portfolio weights** | | | | |
| | Mean | St Dev | Min | Max |
| FAANG/GOLD | 0.14 | 0.09 | 0.00 | 0.55 |
| FAANG/SHARIAH | 0.23 | 0.14 | 0.00 | 0.75 |
| GOLD/SHARIAH | 0.56 | 0.16 | 0.16 | 1.00 |

The table presents the hedge ratios (long/short) and portfolio weights between FAANG stocks, gold, and shariah-compliant stocks.

### 4.3. Hedge Ratios

Theoretically, taking a long position in one asset (say FAANG stocks) can be hedged by talking a short position in a second asset (say gold). Following Kroner and Sultan (1993) and utilizing the conditional variances retrieved from the DCC procedure, we constructed the hedge ratios as follows:

$$\beta_{ij,t} = \frac{h_{ij,t}}{h_{ii,t}} \tag{6}$$

The mean values of the hedge ratio between the three assets are reported in Table 4. The mean value of the hedge ratio among FAANG and gold was 0.49 whereas the mean value of the hedge ratio amongst FAANG and shariah-compliant equity was 0.02, while the mean value of the hedge ratio between shariah-compliant equity and gold is 0.12. The implications of these results are vital to construct an optimal portfolio from a risk management perspective. For instance, as per our results, a USD 1 long position in FAANG can be hedged for USD 0.49, with a short position in the gold market, whereas a USD 1 long position in FAANG can be hedged for just USD 0.02, with a short position in the shariah-compliant equity; in contrast, a USD 1 long position in shariah-compliant equity can be hedged for USD 0.12 with a short position in gold. The most expensive hedge is long FAANG and short gold, while the least expensive hedge is long FAANG and short shariah-compliant equity.

## 5. Summary and Conclusions

The purpose of this paper was to investigate the volatility spillovers and conditional correlations among the daily returns of FAANG company stocks, gold, and shariah-compliant equities. We also constructed the optimal portfolio weights and hedge ratios during the COVID-19 pandemic period. We used the VAR–GARCH framework to estimate the volatility spillovers. The DCC framework was adopted to analyze the time-varying correlations among the three asset classes, which was instrumental in finding the optimal portfolios and hedge ratios among the three indices. The findings indicated the significance of the DCC model while estimating the time-varying correlations as opposed to using static correlations. Using the findings from the DCC model, it emerged that FAANG stocks, gold, and sharia-compliant equities exhibited lower correlations during times of market volatility brought about by recession fears. Therefore, the findings suggest that

gold and sharia-compliant stocks can hedge stocks during periods of market downturn. These findings are highly relevant for international investors, asset managers, hedgers, and portfolio managers. The scope of the current study was limited to only three classes of assets; however, the study can be further extended by adding more classes of assets, and instead of a global investor prospect, a future study may explore regional diversification opportunities.

**Author Contributions:** Conceptualization, K.S.; Methodology, K S. and O.F.; Formal analysis, K.S. and H.K.; Data curation, O.A.; Writing—original draft, K.S. All authors have read and agreed to the published version of the manuscript.

**Funding:** This research received no external funding.

**Data Availability Statement:** Not applicable.

**Conflicts of Interest:** The authors declare no conflict of interest.

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
