# Peer review of "FAANG Stocks, Gold, and Islamic Equity: Implications for Portfolio Management during COVID-19"

_risks, doi:10.3390/risks11010019_

Round 1

Reviewer 1 Report

Although the article is about one of the most current challenges, no references from 2021 or 2022 has been cited and reviewed in the manuscript. Reviewing and including recent references are critical and necessary for assessing the validity and novelty of this article.  Some of the recent studies that I recommend the authors to review are mentioned below:

- Ali, Fahad, et al. "An examination of whether gold-backed Islamic cryptocurrencies are safe havens for international Islamic equity markets." Research in International Business and Finance 63 (2022): 101768. - Yousaf, Imran, and Larisa Yarovaya. "Spillovers between the Islamic gold-backed cryptocurrencies and equity markets during the COVID-19: A sectorial analysis." Pacific-Basin Finance Journal 71 (2022): 101705. - TamošaitienÄ—, Jolanta, Vahidreza Yousefi, and Hamed Tabasi. "Project Portfolio Construction Using Extreme Value Theory." Sustainability 13.2 (2021): 855. - Mirza, Nawazish, et al. "The resilience of Islamic equity funds during COVID-19: Evidence from risk adjusted performance, investment styles and volatility timing." International Review of Economics & Finance 77 (2022): 276-295. - Ashraf, Dawood, Muhammad Suhail Rizwan, and Ghufran Ahmad. "Islamic equity investments and the COVID-19 pandemic." Pacific-Basin Finance Journal 73 (2022): 101765.

Author Response

We have significantly improved the introduction section with the help of referee’s suggested literature.

Reviewer 2 Report

Review Report

Journal: Risks

Manuscript ID: risks-2010926

Overview

The authors set out to investigate the volatility spillovers and dynamic conditional correlations among the daily returns of FAANG company stocks, gold, and Sharia-compliant equity to construct the optimal portfolio weights and hedge ratios during the COVID-19 pandemic period. They find that Gold and Shariah-compliant equity, both are good candidates to hedge FAANG stocks.

General comment

The study is interesting but lacks several important elements to warrant an outright publication. The paper falls in line with several strands of literature but the manuscript fails to synthesize the existing works. I encourage the authors to consider the comments below to improve the study’s contribution.

Main issues

1.      The abstract can be updated to capture the relevance of the study, particularly why FAANG stocks.

2.      In the Introduction, the discussion on COVID-19’s impact on stock markets could be improved and updated in line with more recent (i.e., those in 2022) studies. E.g., https://doi.org/10.1016/j.gfj.2021.100653, https://doi.org/https://doi.org/10.1016/j.iref.2021.09.019, https://doi.org/10.1080/1331677X.2021.1910532, https://doi.org/10.1155/2021/4917051, https://doi.org/10.1016/J.HELIYON.2022.E09215. These are only suggestions, that could help situate the current study into the existing literature, and not forced on the authors. As it stands, the authors limit themselves to papers published in 2020. Updating the literature is important to improve the marginal contribution of the paper.

3.      The methods employed should be compared to existing ones. In this respect, the drawbacks of the methods employed should be discussed.

4.      I recommend that a separate literature review section is presented. The current study belongs to several strands of literature. Hence, a separate literature review will be relevant to know the level of existing knowledge and how the current study contributes to such knowledge.

5.      The sample period (January-September 2020) is too short (less than a year out of almost 3 years into the pandemic). The authors should note that the situation in world markets has changed. Despite the fact that the pandemic has not been dealt with entirely, the Russia-Ukraine conflict has brought additional complications. Why does the sample period not extend to recent times of the pandemic, and possibly the geopolitical risk era?

6.      The authors should relate the discussion of their results to the existing literature that discusses a similar theme, i.e., covering the impact of the pandemic, safe-havens, and tech-based investments.

7.      In addition, the implications of the results should be discussed.

8.      It will be better to suggest a way forward for future studies. This can be done properly only when the authors have synthesized the existing literature properly.

Other issues

1.      Abbreviations such as FAANG should be written in full on first use.

2.      Please, add explanatory notes to all tables and figures so that the reader can understand the meaning of the table without going through the paper text.

3.      A final proofread of the paper will be of merit.

Author Response

File Attached

Reviewer 3 Report

The paper is appropriate for the journal but there are some issues that need attention.

1. please mention in the introduction section the practical and theoretical use of your results. how do they contribute in terms of risk management or portfolio management to know your good results? How will U.S., Euro, or global traders or investors will benefit from them?

2. In the literature review mention how will the results contribute to or refute previous ones. Do the Islamic banking and the securities that you test have different behavior in terms of volatility models that you will highlight with your results?

3. Please highlight the real contribution of your nice work to other researchers, so your paper will have the paper attention and citation.

5. Enhance your literature with the one of Markov-Swicthing (MS) models. Sometimes the persistence of the ARCH term is due to structural breaks, explained through MS or MS-GARCH models. Please talk about about your test. Explain Why are you using single-regime GARCH models to test your relationship and add the next suggested references in your literature review (it is necessary to include them):

Yan, L.; Garcia, P. Portfolio investment: Are commodities useful? J. Commod. Mark. 20178, 43–55. 

Zheng, T.; Zuo, H. Reexamining the time-varying volatility spillover effects: A Markov switching causality approach. North Am. J. Econ. Financ. 201326, 643–662

Balcilar, M.; Demirer, R.; Hammoudeh, S. Investor herds and regime-switching: Evidence from Gulf Arab stock markets. J. Int. Financ. Mark. Institutions Money 201323, 295–321

- https://doi.org/10.3390/math9091030

- https://doi.org/10.1016/j.najef.2013.05.001

- https://doi.org/10.1016/j.ijforecast.2018.05.004

4. Please mention your work limitations and give guidelines for further research in your conclusions.

I hope that these suggestions will improve your nice work.

Author Response

File attached

Round 2

Reviewer 2 Report

Review Report

Journal: Risks

Manuscript ID: risks-2010926

The authors have addressed some of the issues while others remain partially or fully unresolved. I recommend that the authors pay attention to the following.

1.      The authors should update the abstract to capture the relevance of the study, particularly why FAANG stocks. A sentence or two on the importance of the purported theme would suffice. The revised version does not capture this.

2.      There are still language issues that need to be resolved by the authors. E.g., "... number of factor." on line 42; "... based on assumption that ..." on line 45; "... to the examine..." on line 88; "This scope of current study..." on line 273, etc.

3.      The methods employed should be compared to existing ones. In this respect, the drawbacks of the methods employed should be discussed. These issues have not been addressed in the revised manuscript. The use of the approach by several studies is not a convincing reason; it only makes this study an econometric exercise, which should not be so. The authors should explain, in the manuscript, what features or attributes of the methodology make it the most appropriate relative to other approaches.

4.      Again, the authors fail to discuss the implications of the findings in the empirical results section. How can the findings benefit investors, portfolio managers, policymakers, etc? This is yet to be addressed.

5.      Please, add explanatory notes to all tables and figures so that the reader can understand the meaning of all tables and figures without reference to the main text. Not all Tables and Figures have explanatory notes in the revised manuscript.

6.      A careful proofread of the manuscript is again recommended.

Author Response

File Attached 

Round 3

Reviewer 2 Report

Accept.